# Microneedles in Advanced Microfluidic Systems: A Systematic Review throughout Lab and Organ-on-a-Chip Applications

**DOI:** 10.3390/pharmaceutics15030792

**Published:** 2023-02-28

**Authors:** Renata Maia, Violeta Carvalho, Rui Lima, Graça Minas, Raquel O. Rodrigues

**Affiliations:** 1Center for MicroElectromechanical Systems (CMEMS-UMinho), University of Minho, Campus de Azurém, 4800-058 Guimarães, Portugal; 2LABBELS—Associate Laboratory, 4806-909 Braga/Guimarães, Portugal; 3ALGORITMI Center, University of Minho, Campus de Azurém, 4800-058 Guimarães, Portugal; 4MEtRICs, Mechanical Engineering Department, University of Minho, Campus de Azurém, 4800-058 Guimarães, Portugal; 5CEFT—Transport Phenomena Research Center, Faculty of Engineering, University of Porto, Rua Dr. Roberto Frias, 4200-465 Porto, Portugal; 6ALiCE—Associate Laboratory in Chemical Engineering, Faculty of Engineering, University of Porto, Rua Dr. Roberto Frias, 4200-465 Porto, Portugal; 7Advanced (Magnetic) Theranostic Nanostructures Lab, Nanomedicine Unit, INL–International Iberian Nanotechnology Laboratory, Av. Mestre José Veiga, 4715-330 Braga, Portugal; 8Division of Engineering in Medicine, Brigham and Women’s Hospital, Department of Medicine, Harvard Medical School, Cambridge, MA 02139, USA

**Keywords:** microfluidic, microneedles, organ-on-a-chip, lab-on-a-chip, drug screening, biomarkers detection

## Abstract

Microneedles (MNs) have been widely used in biomedical applications for drug delivery and biomarker detection purposes. Furthermore, MNs can also be used as a stand-alone tool to be combined with microfluidic devices. For that purpose, lab- or organ-on-a-chip are being developed. This systematic review aims to summarize the most recent progress in these emerging systems, to identify their advantages and limitations, and discuss promising potential applications of MNs in microfluidics. Therefore, three databases were used to search papers of interest, and their selection was made following the guidelines for systematic reviews proposed by PRISMA. In the selected studies, the MNs type, fabrication strategy, materials, and function/application were evaluated. The literature reviewed showed that although the use of MNs for lab-on-a-chip has been more explored than for organ-on-a-chip, some recent studies have explored this applicability with great potential for the monitoring of organ models. Overall, it is shown that the presence of MNs in advanced microfluidic devices can simplify drug delivery and microinjection, as well as fluid extraction for biomarker detection by using integrated biosensors, which is a promising tool to precisely monitor, in real-time, different kinds of biomarkers in lab- and organ-on-a-chip platforms.

## 1. Introduction

Microfluidic technology is present in lab-on-a-chip and organ-on-a-chip platforms. In order to enable high-throughput screening and automation, lab-on-a-chip (LoC) devices—also known as multitasking devices—combine many (bio)chemical laboratory operations in a single integrated chip that ranges in size from a few millimeters to a few square centimeters [1]. The most alluring benefits of these platforms are their capacity to autonomously and efficiently execute a number of lab processes on a single chip with minimal external inputs [2], as well as with low reagent consumption and high-throughput analysis [3]. To offer an in-situ and quick result for an immediate diagnosis and treatment, point-of-care testing (POCT) is required. For modern POCT diagnostic systems, sample-to-answer format, high sensitivity, and a short analysis time are the most crucial qualities. Since LoC can miniaturize and combine the majority of the functional modules used in central labs into a tiny chip, LoC technologies have been regarded as one of the potential options that can satisfy the needs of POCT [4]. For example, Samper et al., 2019, describe a 3D printed chip to create a microfluidic biosensing portable system, where the data is transmitted via Bluetooth [5]. Another example is the study of Zhang et al., 2020, which demonstrated the integration of a smartphone detection into a microfluidic device (acoustofluidic platform) for hemoglobin measurement. To detect the fluorescent signal, the researchers created a quantum dot-based fluorescence test for hemoglobin and paired it with an integrated UV irradiation source and a commercial smartphone [6].

Organ-on-a-chip (OoC) platforms replicate tissue and miniaturized organs, while preserving tissue/organ-level function and homeostasis [7]. They are found on microfluidic devices with perfused chambers that range from micrometers to millimeters in size, and are fed by continuous media flow [8]. As a result of the continuous flow of cellular media, shear flow conditions and nutrient/gas exchanges, OoC can be mimicked as in vivo, extending the cell culture’s lifetime compared to static in vitro cultures [9]. Therefore, OoC can reproduce important features of the complexity of organs and biosystems [10]. Several studies in the literature use OoC to examine specific target organs, including the liver [11], heart [12], brain [13], and kidneys [14], among others. The aim of many of the OoC is to facilitate drug toxicity detection in healthy and diseased organ models. Because OoC can include patient primary human cells or stem-cell-derived cells, the OoC system has the potential to be designed as a model platform capable of predicting optimized and personalized drug treatments [15]. However, important hurdles must be overcome to create a valid and robust preclinical organ model. For that, appropriate organ scaling, tissue vascularization, recapitulation of the immunological response, repeatability, organ monitoring, oxygenation, pH, shear rate, cell viability, and cell density, are some of the parameters that need to be considered when designing an OoC [16]. Among all these features, monitoring the OoC platforms is a huge step to guarantee reproducibility and appropriate chemical, physical, and cell analysis. Therefore, OoC and LoC can be combined, especially regarding the integration of micro (bio)sensors of LoC into OoC, bringing advanced microfluidic devices into a new era.

Microneedles (MNs), which are based on the concept of miniaturized needles, are increasingly used in biomedical technology. These have the ability to assess biological information with minimal invasion, and are frequently used as a strategy to deliver drugs [17], biomolecules such as proteins [18], RNA, or DNA [19] into cells with temporal and spatial precision [20,21]. The dimensions of MNs may vary depending on the application. The most common dimensions found in the literature have height ranges between 150 to 1500 μm, with a base width of 50 to 250 μm and a tip diameter of 1 to 25 μm [22]. In terms of shape, needle tips come in a variety of shapes, including triangular, cylindrical, and pentagonal [23]. The design and size of MNs have been identified as the primary characteristics to be modified for optimal performance of an MNs system. To maximize efficiency, the length of the MN can be customized to achieve the desired depth of penetration. The shape, the number of needles in an array, the height, the aspect ratio (the ratio of the base to the height of the needle), the material, and the thickness of the backing block (if needed), are all criteria that define MN design. In addition, the volume that can be collected and loaded by the array is determined by these criteria. The volume, in turn, contributes to determining the type of MN that best suits the desired application [24]. Based on applications, MNs can be categorized into various types. MNs systems have mostly been developed for biomolecular/drug delivery and microinjection [25,26,27,28,29,30,31,32]. The design of the MNs device is crucial for the efficient performance of the system, and different materials can be used in MNs fabrication [33]. The two fundamental designs that are employed to construct MNs are in-plane and out-of-plane (Figure 1A). In contrast to out-of-plane MN arrays, which rise vertically from the base, in-plane MN arrays are parallel to the top fabrication surface [34]. Typically, due to the numerous microstructures and variety of strategies for the delivery of drugs, MNs are divided into two main categories: traditional needles (solid, coated, or hollow), and emerged needles (dissolving, hydrogel-forming) [35]. In terms of materials, MNs can be divided into degradable and non-degradable materials, such as metal, silicon, ceramic and carbon for non-degradable and natural polymers for degradable ones [36,37]. Figure 1B represents the two main categories with the approaches of the six most used MNs. A more comprehensive review of these MNs structural strategies can be found elsewhere [38].

Concerning the fabrication methods, several have been described in the literature, but the most commonly used are micro-molding, microfabrication technologies (e.g., lithography, laser, etching), additive manufacturing (i.e., 3D printing), and layer-by-layer assembly [39]. Briefly, microfabrication can also be divided into three main processes: deposition, patterning, and etching. Deposition includes film formation by physical vapor deposition or chemical vapor deposition. The patterning technique shapes the desired geometry on a film, substrate, or wafer. Lithography is a common technique used for patterning, which consists in transferring the mask into a coated photosensitive film using light to develop the exposed photoresist. Although lithography allows the production of smaller feature sizes, it is considered a more complex process that requires high-tech infrastructures and equipment [40]. Etching is a technique that involves removing the unprotected sections of the substrate with a strong caustic chemical to create a microneedle design of interest. A wet or dry etching technique can be used, but the use of chemicals are required, which can contaminate the samples [41]. Laser ablation and laser cutting are also reported to be used to fabricate metal and polymeric MNs. Laser ablation removes material from a solid surface by irradiating it with a laser beam [42]. Laser cutting uses an infra-red laser to cut metallic sheets in the shape of MNs [43]. Both techniques are simple, quick and precise, with no contaminations, but require higher power consumption. Micro-molding is used to fabricate various polymeric MNs using cutting tools to sculpt the mold. Afterwards, the polymeric material that comprises the MN is poured into the micro-mold in a liquid or semi-liquid state and then solidified to achieve the desired shape. It is a simple, low-cost, versatile process with high-resolution [44,45]. More recently, 3D printing has also emerged as a process to produce MNs with the potential to simplify the fabrication of multilayer and materials in a few steps [46].

Overall, the microfabrication techniques to produce MNs and microfluidic devices are identical. Hence, it is expected that microfluidic devices and MNs can be easily combined using those fabrication techniques and in this way to create, in a synergetic way, an advanced microfluidic device for drug screening and/or organ models monitoring [47,48]. Based on this expectation, the present systematic review aims to provide a broad vision on the state-of-the-art of MNs combined with lab and/or organ-on-a-chip, especially focusing on the MNs type, fabrication strategy, materials, and function/applications.

## 2. Materials and Methods

This work was conducted taking into account the research guidelines for systematic reviews proposed by PRISMA [49,50].

### 2.1. Data Sources and Search Strategy

The search was performed using three different databases: ScienceDirect, PubMed and Scopus, until the 1st of December 2022. The search string used was (“organoids” OR “organ-on-a-chip” OR “organ on a chip”) AND (“microneedle (s)”) AND (”lab-on-a-chip” OR “microfluidics” OR lab on a chip”) AND (“microneedle (s)”).

### 2.2. Validity Assessment

Review articles, conference papers, short communications, and non-English written articles were removed from the search results, either manually or using the filters from the database. After the elimination of duplicates, the articles were selected based on the relevance of their title in the context of this review. Further screening was performed to evaluate which paper presented the defined inclusion or exclusion criteria presented below. To avoid biases, the two first authors screened and selected the research papers separately and then compared the classifications. Disagreements or doubts regarding the classification were solved by a third author.

### 2.3. Inclusion and Exclusion Criteira

The studies included in this review followed the criteria: Published since 2000;Use of microfluidic platforms or organs-on-a-chip in combination with MNs;Use of microfluidic platforms or lab-on-a-chip in combination with MNs;MNs for media/ISF collection;MNs for cell injection;MNs for biomarkers detection.MNs for biofluid extraction, microneedle sensors, and analyte-capturing MNs, or combinations thereof.

The study did not present the excluded criteria.

## 3. Results

### 3.1. Data Collection Results

As previously mentioned, the authors followed the PRISMA-recommended guidance to conduct systematic reviews. Based on the title, 91 potentially relevant articles were identified from the three databases selected. In total, 80 studies were included after removing duplicates. After the evaluation of abstracts, 24 articles were dismissed due to a lack of data and different study strategies; thus, 56 full papers were analyzed. In the end, a total of 35 full-text articles were selected. Figure 2 shows the PRISMA flow chart for the selection process of studies incorporated in this systematic review.

Additionally, a metadata analysis was carried out using the Scopus database with the searched keywords “MNs + microfluidic” and “MNs + organ-on-a-chip” and “MNs + lab-on-a-chip” between 2000 (the year of the first work reported in the literature) and 2022, which shows a total sum in this period of 82 papers (67 articles and 15 reviews) (Figure 3).

Among the included studies, 30% corresponded to studies that have addressed the integration of MNs into microfluidic devices. The number of publications tended to increase over the past 20 years, where the majority of the publications were original research articles. This reflects researchers’ increased interest in combining MNs with lab/organ-on-a-chip. Among the most published areas were Engineering, Material Sciences, and Physics and Astronomy, representing a total of 63.4%. 

Based on the selected articles from the defined criteria, 36 works were included in this systematic revision, which had as main topic MNs applied in advanced microfluidic devices (i.e., lab/organ-on-a-chip), and subdivided into two main applications: (1) devices for extraction and biomarker detection, and (2) devices for drug delivery and microinjection, as follows in the sub-chapter.

### 3.2. MNs Applied in Advanced Microfluidic Devices

An increased research effort has been focused on the use of MNs for direct or indirect sensing. This new trend has germinated naturally from former efforts of the use of MNs for OoC and LoC devices. As is possible to observe in Table 1, the majority of the applications of MNs in the microfluidic field are LoC approaches (approximately 89% of papers analyzed). Generally, an MN array is connected to a reservoir to serve as an interstitial fluid (ISF) absorption device or connected with a reservoir to serve as drug storage to release drugs.

Hollow MNs are regarded as the best choice for extract/release systems, since they provide the exact amount of drug needed at the desired location in a faster and controlled way. Therefore, hollow MNs are the most common type of MN employed in microfluidic devices, mostly made of silicon, but metals and glass are also used (Table 1). Such devices act as a conduit to access dermal biofluids for on-chip analysis in microfluidic chambers [51]. However, besides hollow MNs, the application of porous and solid MNs in microfluidic devices are also established (Table 1).

To identify the presence of a particular target analyte, microneedle-based biofluid extraction products are mainly combined with downstream analytical techniques [52]. For example, Wang et al., 2021 explored an MN patch for fast in vivo sampling and on-needle quantification of target protein biomarkers [53]. Microneedle-based in vivo sensors have been used in diagnostic systems functioning as electrodes, particularly for glucose testing [54]. To overcome the gap between extracting ISF and further analysis, some authors proposed solutions that incorporate the biosensor on the patch of porous MNs [55]. For example, Kusama et al., 2021, proposed a porous MNs patch combined with anodes and cathodes for efficient drug delivery (penetration) and analysis (extraction) [56].

Ultimately, MNs systems can provide results and detect different biomarkers in real-time, which can be used to monitor in vivo tissues, or in vitro organoids and cell cultures. The main advantage of MNs is that they can be repeatedly used to collect cellular contents without causing cell lysis. They may also promote a decrease in lateral diffusion [57]. 

Overall, studies show that MNs are mainly used in microfluidic applications for biomarker detection [58,59,60,61,62,63,64,65,66,67,68], cargo delivery [69,70,71,72,73], and cell microinjection [74,75]. Table 1 shows the types of MNs, materials, applications and hydrodynamic forces used in microfluidic devices. The works reviewed show that MNs are designed in two configurations, in-plane and out-of-plane (as shown in Figure 1A). An in-plane MN configuration enables the manipulation of the length and shape of MNs and the time required to produce it. It simplifies the integration into an embedded microfluidic network resulting in a device with fewer layers and steps process. Parameters, such as mechanical rigidity, can be easily tailored by varying the subtract thickness or width of MNs [76,77,78]. As a result, these MNs are often longer than out-of-plane MNs [79]. Out-of-plane MNs can also enhance the efficiency of drug delivery/fluid extraction by increasing the MN array density. However, achieving a higher length is more difficult because of the risk of clogging and collapsing [58,80,81].

Another interesting aspect in the design and application of MNs in microfluidic devices is the type of hydrodynamic force mechanism employed, which can be passive, such as capillary force, or active, such as by using micropumps. For instance, the detection of analytes in fluids may be facilitated by the use of capillary action in a microneedle-assisted biosensing [82]. In this case, capillary forces can propel the fluid to/from the reservoir and then to a biosensor platform. However, in some cases a micropump can be requested to supply specified volumes at higher flow rates, which in turn adds more complexity to the system [83]. Nevertheless, in many designs, capillary forces are enough and allow the liquid to flow through the MNs on its own, simplifying the manufacture and use of the device [84]. When natural hydrodynamic forces, such as capillary, are not enough, other components such as pumps, valves, and bubble traps must be combined in order to achieve the system. This, as already mentioned, can be challenging from a fabrication and integration standpoint [59]. In some revised works, planar micropumps were used because of their advantage in being simple to integrate and having the ability to change the flow operation (extraction/injection) by just flipping the valve direction. For LoC devices with in-plane MNs, this approach is further explored [60,70], although leaks between the inlet and outlet can occur. 

Regarding fabrication strategies and materials, typically porous MNs are fabricated using a PDMS mold followed by a leaching method to remove the porogenic casted materials [61,62], or by using a microfabrication process to directly obtain the MNs’ structure, followed by leaching [64]. The majority of solid MNs are built of metallic components or silicon, or a combination of both [65,73,85]. In the studied papers, the solid MNs were produced through micromachining processes, including the use of SU-8 photoresist. On the other hand, coated MNs are in general solid MNs that suffer a process of coating. For example, Trzebinski et al., 2012, developed a microfluidic device with enzyme-coated MNs by immersing the MNs in a solution with the desired enzyme [67]. Kang et al., 2021 used a silicon-coated MN with Cr/Au by deposition [71]. In contrast, different types of lithography are commonly used in the case of hollow MNs [63,86,87]. Deep reactive ion etching (DRIE) and sacrificial layer sharpening are two other techniques that have been extensively researched in MNs and used in the investigated microfluidic devices [70,74,75,76,77,78,81]. New fabrication processes, such as 3D printing, are starting to be developed as well. A comprehensive review concerning this fabrication methodology for the design of MN for biomedical application can be found elsewhere [88].

**Table 1 pharmaceutics-15-00792-t001:** Reviewed MNs-based systems concerning MN type, MN–chip connection, fabrication strategy, material, employed hydrodynamic force, function, application and microfluidic system.

**MN** **Type**	**In/Out-of-Plane**	**MN–** **Chip Connection**	**Fabrication** **Strategy**	**Material Chip/MN**	**Forces**	**Function**	**Application**	**Microfluidic** **System**	**Reference**
Porous MNs	Out-of-plane	MN integrated in the inlets of the microdevice	Microfabrication + Leach method	Polylactic acid (PLA)/PDMS	Pump	Biomarker detection	ISF collection and glucose detection	Lab-on-a-chip	[61,62]
Porous MNs	Out-of-plane	Integrated as MN patch	Mold + Leach method	PDMS/Ethoxylated trimethylolpropane triacrylate (ETPTA)	Capillary Action	Biomarker detection	Extraction and detection of skin interstitial fluid biomarkers	Lab-on-a-chip	[64]
Solid MNs	Out-of-plane	MN integrated in the inlets of the microdevice	Microfabrication (SU-8)	PDMS/SU-8	Pressure	Drug delivery	Delivery functions for inflammation treatment	Lab-on-a-chip	[73]
Solid MNs	In-of-plane	MN integrated perpendicular to the microfluidic channel	Microfabrication	Oxide layer + metallic layer/Silicon	-	Biomarker Detection	Microneedle biosensor for direct label-free real-time protein detection	Lab-on-a-chip	[65]
Solid MNs	Out-of-plane	MN integrated perpendicular to the microfluidic channel	-	PDMS/Tungsten + parylene	Syringe pump	Cell Detection	Detection of cells in suspension	Lab-on-a-chip	[85]
Coated MNs	Out-of-plane	MN integrated above microfluidic channel	Microfabrication	PDMS/Silicon + Cr/AU	Capillary Forces	Delivery	Chemical delivery capability	Lab-on-a-chip	[71]
Coated MNs	Out-of-plane	MN integrated above chamber	Microfabrication (two-photon lithography)	PDMS/Gold + enzyme layer	Syringe pump	Biomarker detection/Biosensor	3D microspike array-based glucose and lactate biosensor	Lab-on-a-chip	[67]
Coated MNs	Out-of-plane	MN integrated above microfluidic channel	SU-8	PDMS/SU-8 resin	Syringe pump	Biomarker detection/Biosensor	Drug delivery and body fluid sampling applications	Lab-on-a-chip	[68]
Hollow MNs	Out-of-plane	MN integrated above microfluidic channel	Microfabrication	-	Micropump	Biomarker detection	Nonenzymatic microfluidic glucose sensor	Lab-on-a-chip	[60]
Hollow MNs	Out-of-plane	MN integrated in organoid chamber	Microfabrication (Photolithography)	PMMA/Silicon	Pneumatic interface	Biomarker detection	Microfluidic sampling system for tissue analytics	Organ-on-a-chip	[87]
Hollow MNs	Out-of-plane	MN integrated above microfluidic channel	Microfabrication	Pyrex/Silicon	Capillary action and evaporation	Biomarkerdetection	Microneedle-based glucose monitor	Lab-on-a-chip	[58]
Hollow MNs	Out-of-plane	MN integrated above microfluidic channel	Microfabrication/DRIE	Aluminum + Silicon/Silicon	Capillary forces	Extraction	ISF extraction	Lab-on-a-chip	[81]
Hollow MNs	Out-of-plane	MN integrated above microfluidic channel	Microfabrication (two-photon lithography)	PDMS/Eshell 300	Pump	Analysis	Sensor for on-chip potentiometric determination of K^+^	Lab-on-a-chip	[63]
Hollow MNs	Out-of-plane	MN integrated perpendicular to the microfluidic channel	Soft lithography	PDMS + SU-8/Glass	Valve actuation	Micro-injection	Single cells microinjection system	Organ-on-a-chip	[74]
Hollow MNs	Out-of-plane	MN integrated above microfluidic channel	Direct laser writing	PMMA/Photosensitive material	Syringe	Extraction/ delivery	A system for fluid injection and extraction	Lab-on-a-chip	[86]
Hollow and sharp MNs	Out-of-plane	MN integrated above microfluidic channel	Laser Ablation	Glass/SU-8	Syringe pump	Perfusion	3D micro perfusion system	Organ-on-a-chip	[89]
Hollow MNs	Out-of-plane	Integrated as MN patch	Soft lithography	PDMS/metal	Pressure	Extraction	Extraction and transport of blood	Lab-on-a-chip	[90]
Hollow MNs	Out-of-plane	Integrated as MN patch	Soft lithography	PDMS + paper sensor	Pressure	Biomarker detection	POCT biosensors for quantification of glucose and cholesterol in blood	Lab-on-a-chip	[66]
Hollow MNs	Out-of-plane	MN integrated perpendicular to the microfluidic channel	3D printing + DRIE	PDMS/Glass	Vacuum pump	Microinjection	Microfluidic device for localized microinjection	Lab-on-a-chip	[75]
MN with open capillary	In-plane	Connected with microfluidic device	DRIE + photolithography	Silicon	Pressure	Insertion into skin	Extraction/delivery	Lab-on-a-chip	[76,77]
MN with open capillary	In-plane	Connected with microfluidic device	DRIE + photolithography	Titanium	Pressure	Insertion into skin	Extraction/delivery	Lab-on-a-chip	[78]
MN with open capillary	In-plane	Connected with microfluidic device	MEMS + glass cover on silicon technology	Silicon	Syring pump	Drug Infusion	System for brain drug infusion	Lab-on-a-chip/organ-on-a-chip	[72]
MN with open capillary	In-plane	Connected with microfluidic device	MEMS + DRIE	Silicon	Planar Micropump	Drug Delivery	Continuous on-chip micropumping for microneedle enhanced drug delivery	Lab-on-a-chip	[70]

#### 3.2.1. Devices for Extraction and Biomarker Detection

The use of microfluidic components combined with MNs for glucose measurement has received the greatest attention [67,91]. As is possible to observe in Table 1, and as mentioned above, hollow MNs are the most developed for fluid extraction that serve for biomarker detection. In general, such devices act as a conduit to access dermal biofluids for on-chip analysis. Historically, the first LoC devices were designed with hollow MNs [58,81]. For instance, Mukerjee et al., 2004 integrated a hollow MN array with microchannels to measure glucose levels in situ. The fabrication process was a combination of DRIE and isotropic etching to produce out-of-plane hollow MNs integrated with microchannels and reservoirs. The chip was designed to draw fluid from the MN tip to microchannels by capillary forces. To achieve it, the surface tension was optimized by studied geometry, contact angle and an MN cross-section, and it was concluded that the best silicon MN profile for extracting fluid was the “snake fang”. The device was able to successfully pierce the skin and extract fluid, and glucose was measured by calorimetry with the fluid in the reservoir (Figure 4A) [81]. However, the study reported that an inflammatory response was observed. Another extraction system was described by Lee et al., 2012, in which the system integrated an ultrahigh-aspect-ratio (UHA) microneedle with a novel elastic self-recovery actuator. This device successfully extracted and transported blood from a rabbit [90]. In addition, biosensors started to be combined into the microfluidic system/device with microneedles. As described by Zimmermann et al., 2003, a disposable minimally invasive self-calibrating sensor for continuous glucose monitoring was developed, consisting of hollow out-of-plane MNs to sample ISF from the epidermis that was placed in a shallow flow channel. Capillary action and evaporation drove the ISF through the MNs into the integrated glucose sensor. However, it was a prototype test, and the authors suggested that more investment was needed for the fluid to reach the biosensor. In addition, it was reported that the passing fluid gradually washed away the immobilized enzyme [58]. 

As a possible solution, Najmi et al., 2022, developed and simulated a nonenzymatic glucose detection device by integrating a microfluidic system and a semi-permeable membrane located at the MN base to separate the dialysis fluid from the waste fluid. In this case, an amperometric sensor was used [60]. However, some leakage was observed from outlet-to-inlet (Figure 4B).

Mansor et al., 2017, created a device for the detection of cells in suspension. The microfluidic chip consists of two MNs integrated at both sides of the channel to detect impedance measurements of passing cells through the applied electric field. The MNs can be reused, but each PDMS microchip can only be used for one cycle. Due to the low fabrication cost and more than one functionality (solution detection and cell concentration detection), this device was described as suitable for various applications, such as cancer cell detection and water contamination [85] (Figure 4C).

Miller et al., 2014, developed an ion-sensitive microfluidic chip with hollow MNs, one approach that provides an attractive platform for an on-body sensing system for monitoring potassium, which can be easily expanded to other relevant physiological markers for the next generation of point-of-care diagnostic devices. This work was the first ion-selective-electrode MN sensor integrated into a microfluidic device [63] (Figure 4D). A multi-diagnostic system including a PDMS touch-switch, a paper multisensory, and a hollow MN was described by Li et al. 2015 (Figure 4E). The cholesterol and glucose levels in rabbit blood were measured using this device [66]. The main benefit of this method is that it only requires one finger press to activate. With just one finger, enough pressure is applied to the PDMS touch-switch to enable the MN entry into the blood vessel. The deformable chamber returns to its former shape after the finger is released, creating a negative pressure that allows blood to be extracted through the hollow MN and into the sensor-chamber [66]. Sarabi et al., 2021, described an MN array integrated with a microchip for body fluid samples powered by finger press, and the process of fluid flow and its transport across the device was modeled and simulated with a finite model [92].

Although hollow MNs are further explored, some devices incorporate other types of MNs, such as porous MNs. The fundamental benefit of porous MNs is that biodegradable polymers can be used, since their porous structure does not require the micromachining procedures used for hollow structures. Takeuchi et al., 2019, (Figure 4F) developed a microfluidic system with a hydrodynamically designed interface between a porous PDMS MN array and microchannels to enable a direct analysis of liquids extracted by the porous MN array [61]. A porous MN array connected to a microfluidic chip was inserted into agarose gel for evaluation of the collected fluid. This strategy demonstrated a lower flow rate than in the microchip itself, which can be due to the porous MN array increasing the hydraulic resistance of the fluidic connection from the gel to the assay chamber. More recently, a similar system was improved by an additional interface to mechanically and fluidically connect the MN array to microfluidic channels, and tested to show the potential of incorporating MNs in a microfluidic device (Figure 4G) [62]. In another study, Yi et al., 2021, developed a device with porous MNs for the extraction and detection of skin ISF. In this work, a combination of porous MNs with aptamer immobilization was developed, which created an innovative device [64]. In some devices, the detecting method is incorporated in the MN itself, as described by Esfandyarpour et al., 2013, where a solid MN biosensor (with four layers) was developed with the ability to directly measure biomolecular binding as a function of time. This strategy is described as useful for measuring reaction kinetic constants for various biomolecular species [65]. The MNs’ position in this platform differs from the standard presenting horizontal MNs. The vertical construction is described to have the advantage of increasing the transducer sensitivity due to the smaller sensing area, whereas the horizontal form is preferable owing to the simplicity of production (Figure 4H). However, the use of MNs integrated in microfluidic devices can have some limitations for the extraction and detection of biomarkers. Typically, by using these MN-integrated microfluidic systems, a limited volume of sample is collected by unit of time, which can cause a limited detection of biomarkers. This can be particularly challenging for applications that require analysis of biomarkers that are presented in low concentrations, and thus, need more volume to achieve the limit of detection (LOD) in the biosensing unit. Additionally, the accuracy to detect biomarkers in these devices, strongly depends on the specificity and sensitivity of the detection method used. An example is the employment of electrochemical sensors, where their sensitivity can decrease over time due to passing fluid that can wash away the immobilized recognition molecule (antibody/aptamer), or by cleaning steps between readings. Therefore, these limitations are important parameters that should to be taken into account for this application.

#### 3.2.2. Devices for Drug Delivery and Microinjection

Microfluidic devices integrated with MNs can be used for drug delivery due to the precise control of drugs released by means of microfluidic components, such as micropumps. This strategy enhances continuous-on-chip drug delivery. Zhan et al., 2004, describe a delivery system with in-plane MNs with a microchannel and an outlet. This design allows for the decoupling of the mechanical and fluidic performance of the device (Figure 5A) [70]. For example, by using thicker substrates and/or larger shanks, needle shank stiffness can be easily raised without compromising flow rate or inlet pressure. Similarly, by employing larger substrates and more deeply etched channels, the flow rate can be enhanced without increasing inlet pressure or lowering stiffness. Based on this design, but with a different method of fabrication, Lee et al., 2015 proposed and demonstrated a MEMS MNs system for deep brain drug infusion [72].

The first flexible microneedle patch incorporating microfluidic components for on-chip loading and delivery control was produced by Xiang et al., 2015 [73]. The ease of use and cost-effectiveness of the proposed microneedle-fluidic system allow it to be a suitable and promising disposable medical device. Additionally, the device was used in the first in vivo experiment where the local inflammatory phenomena were treated by delivering diclofenac solution transdermally into tissues.

In another study, Kang et al., 2021, fabricated a flexible array base microfluidic neural interface to add a chemical delivery capability to three-dimensional electrode arrays comprising a collection of MNs (a silicon MN coated with Cr/Au) positioned perpendicular to the array base [71]. In this device, the fluid flows along the surfaces of MNs from the base, resulting in fluid delivery directly to the brain surface, but indirectly to the electrodes (Figure 5B). Thus, combining drug/chemical delivery with sensing.

The first microfluidics-based microinjection system was suggested by Adamo et al., 2008, in which single cells were propelled by fluid streams and subsequently injected via MNs with the aid of flexible valve actuation (Figure 5C) [74]. In another strategy, this time based on nanoneedles, Huang et al., 2019 used a microfluidic chip and a nanoneedle array to disrupt the cell membrane, enhancing the passive transfusion of biomolecules to the cytoplasm [93]. Using microinjection-microfluidic systems allows transferring of bioactive agents to many cells in parallel, efficiently transfecting, and works in cell lines difficult to transfect. Zabihihesari et al., 2020, demonstrated the first microfluidic platform that enables immobilization and localized microinjection of a larva due to an integration of a microneedle in the device [75].

As described above, the incorporation of MNs is more developed for LoC devices than for OoC devices. Among the OoC examples is the work of Hokkanen et al., 2015, which developed an automatic microfluidic sample system for tissue analysis. In this work, a microscope and robot were used for positioning the MNs and samples for real-time biopsies (Figure 5D). The blunt-tip silicon MN chips were used to measure indicative cancer biomarkers from the tissues. This system is reported as able to be used to extract lipids from small biopsies [87].

In another OoC study, Choi et al., 2007, described a microfluidic perfusion system that integrates microneedle arrays in a packaged system for fluid containment. The hollow MNs were made with a thin body, a sharp tapered tip, and a microfluidic port along the tower’s side to deliver medium to the inside of the target area (Figure 5E) [89]. The 3D perfusion design provides convective mass transport to the tissue interior for experiments on large tissue preparations over extended time periods.

In-plane MN with an open capillary was another strategy developed for LoC. Jung et al., 2015, fabricated in-plane MNs with side openings with a microchannel inside the MN that connected to a reservoir (Figure 5F) [77]. The fabricated microdevice was described to be applied for minimally invasive drug delivery or sample extraction.

Overall, these papers show the vast potential and versatility of applications for MN in microfluidic devices. However, in general, these devices are made using time-consuming fabrication processes that involve numerous steps and additional sealing layers to enhance the total thickness of the structures. The width of microchannels is frequently constrained by small openings created for successful sealing, making it difficult to reliably seal long microchannels. To overcome this fabrication constraint, Trautmann et al., 2019 described a POCT system combining femtosecond laser-generated microfluidic channels and direct laser-written MN arrays that simplify the fabrication process in only a few processing steps. This is an advantage over multiple processing methods [86]. With this method, hollow MNs of various designs were created, and a flow test using rhodamine B was performed to validate the microchip. Nevertheless, there are some other limitations that must be considered when designing MN-integrated microfluidic platforms for drug delivery and microinjection. For instance, the limited volume of drug that may be administered or injected by unit of time must be considered. This can be especially difficult for applications requiring high therapeutic dosages. In addition, many drug treatments require specific formulations and combinations of formulations, which can pose a challenge for researchers and pharmaceutical companies to adapt to MN administration. This includes using an MN-system applying soluble MNs with specific kinetic drug release. Additionally, for this application, the selected material that will act as MNs must be adequate to not react with the drug formulation. As an example, PDMS is known to capture hydrophilic molecules, which can decrease the release of available drug [94]. Considering the application of MN to be used in OoC or to act as drug delivery in a biological matrix, a challenge that may develop is the MN breaking and clogging during the insertion/extraction of MNs into the biological tissue, which can compromise the functionality of the device. Therefore, the selection and specificity of the material chosen to act as MNs and to be integrated within the microfluidic device is one of the most important parameters in the fabrication of such platforms.

## 4. Conclusions

Microfluidic devices with MNs are a relatively novel and appealing method of fluid transport that offers numerous benefits and applications. Because of the multiple possible applications in the field of biomedicine, the attention to these devices has increased significantly in recent years, as shown by the increasing number of published of lab/organ-on-a-chip systems with MNs. These combined structures are compatible with the current biotech requirements for both operation and assessment, including automated liquid management, plate shuttling, biomarker detection and delivery features. Some of the described LoC devices were tested pre-clinically, which shows the end-use applicability of them. However, systems integration and manufacturing MNs present substantial challenges (e.g., clogging effect, biocompatibility, fluid leakage, numerous fabrication steps and high costs). This inherent complexity can negatively influence the manufacture of such devices’ dependability and repeatability. One of the challenges in the integration of MNs with microfluidic devices is ensuring that the MNs are precisely aligned with the fluidic channels. This is necessary to guarantee that the fluid flows smoothly through the device and the MNs are able to perform their intended function. Another challenge is ensuring that the MNs’ material corresponds to the practical demands of the application that the device is being developed for, as previously outlined in this work.

Overall, the integration of MNs with microfluidic devices is a complex process that requires careful consideration of a number of factors, including design of the MNs, manufacturing techniques, possible functionalization of MN surfaces, and combination or integration with (bio)sensing and actuation systems. As researchers continue to develop new techniques to fabricate MNs-integrated microfluidic devices, the potential application for these devices is expected to expand. As a result, simple fabrication methods and new materials are being explored.

Among MNs, hollow MNs are the ones most employed in microfluidic devices, due to their capability to collect and release higher amounts of fluid. Regarding the MN material, different types of materials from metallic to non-metallic were described in the literature reviewed, which are highly dependent on their application. For instance, MNs designed to penetrate the skin or tissue must be fabricated with materials that provide enough strength and biocompatibility.

Among the microfluidic type of devices, LoC systems with MNs are being explored for extraction, biomarker detection, microinjection, and drug delivery, which can be combined for two or more of these applications in the same device. On the other hand, MN-fluidic systems for OoC/cell-culture monitoring are in the early stages of development. Nevertheless, its potentiality for cell/tissue monitoring is high, as MN structures can measure protein levels directly inside a living cell, without lyse and at real-time. For that, a thin MN can be inserted into a living cell, or in a cluster of cells (tissue), to assess its microenvironment and homeostasis. This can be used for a variety of purposes, including protein expression monitoring, homeostasis assessment, precise drug delivery, and DNA or RNA therapy. Similarly with LoC, one of the most significant advantages of OoC is the versatility of these devices for multiple applications, which has a wide scope for further development. However, the intricate structure of microfluidic devices, as well as challenges in integrating with other devices and fabrication requirements, all contribute to an untapped potential for MNs in the biomedical field. Nevertheless, work is being conducted to overcome this challenge, as shown by a recent published Patent WO/2022/180595, which presents a multiorgan-on-a-chip containing an MN-(bio)sensing platform for the validation and study of nanomaterials, drugs, or mixtures thereof, intended to be used in biomedical and/or pharmaceutic applications [95].

In addition, several researchers are focusing their efforts on developing simpler and low-cost fabrication methodologies to create powerful MN-microfluidic devices, where the integration of the different components, such as biosensing modules, are more easily achieved. One of the newest approaches is the development of microfluidic devices and biosensors using bioprinting technology, which can fabricate in a few steps. This technology, although promising, is also in its first stages. Therefore, there is plenty of room for the MN-microfluidic technology to progress, from material science and microfluidics to tissue engineering. Furthermore, a wide range of applications can be explored, not only biomedical, but also in the sea and in space.

## Figures and Tables

**Figure 1 pharmaceutics-15-00792-f001:**
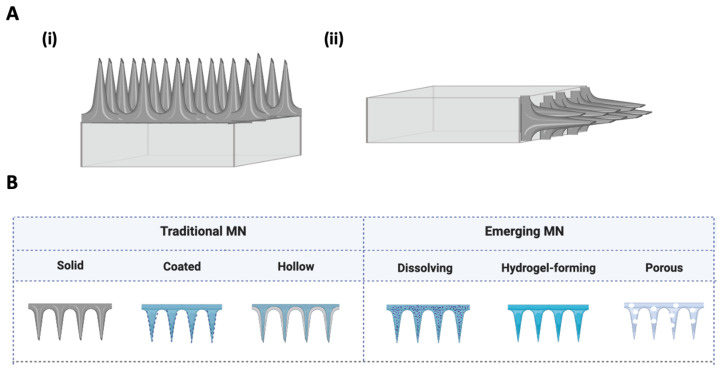
Schematic representation of MNs. (**A**) (**i**) out-of-plane and (**ii**) in-plane construction. (**B**) Traditional and emerging MNs structuring approaches according to drug delivery application.

**Figure 2 pharmaceutics-15-00792-f002:**
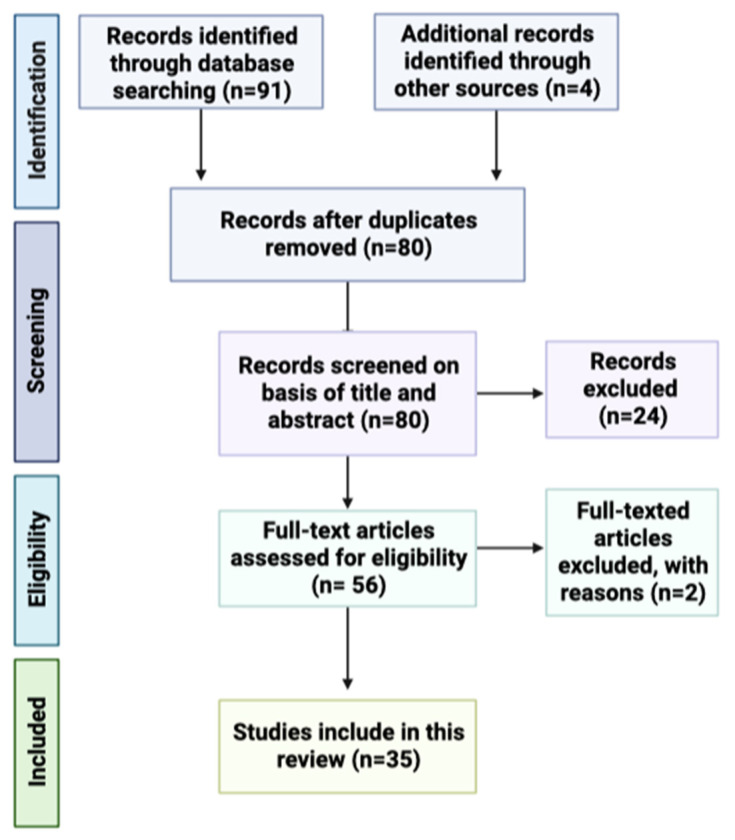
PRISMA flow diagram displaying the procedure of study selection.

**Figure 3 pharmaceutics-15-00792-f003:**
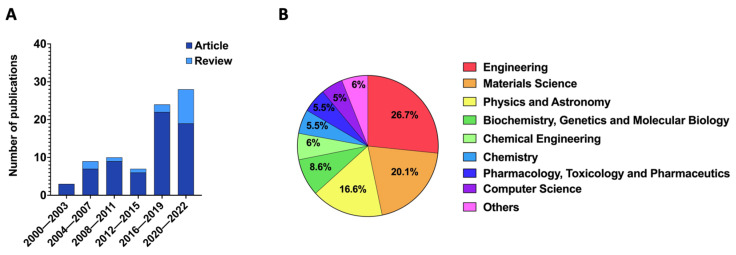
Metadata analysis of the keywords “MNs + microfluidic” and “MNs + organ-on-a-chip” and “MNs + Lab-on-a-chip” between 2000 and 2022. (**A**) Number of publications per combined years. (**B**) Documents by subject area.

**Figure 4 pharmaceutics-15-00792-f004:**
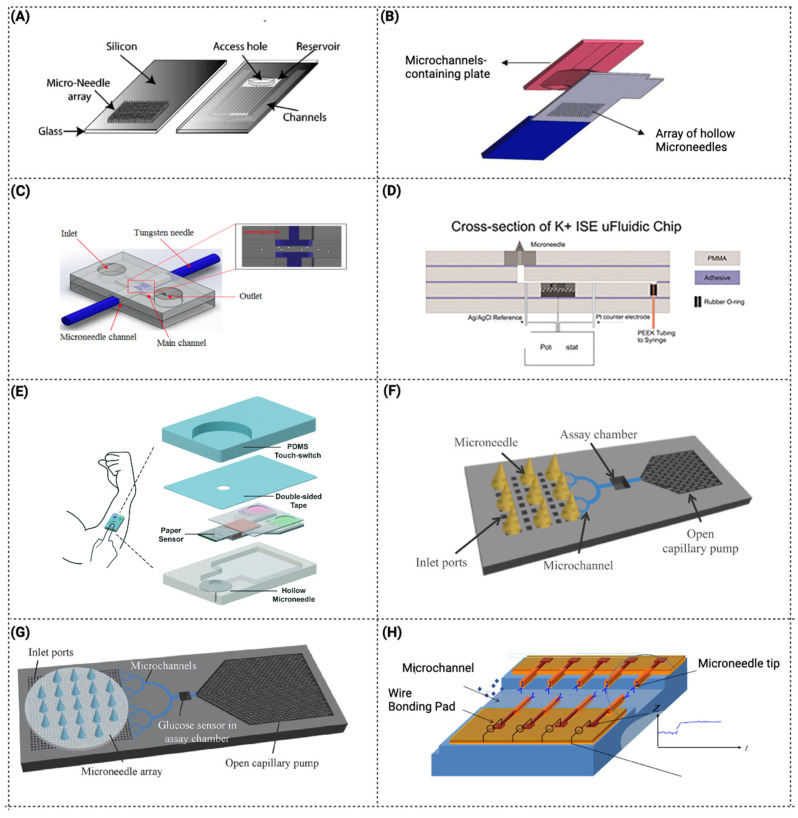
Schematic compilation of MN microfluidic devices for extraction and biomarker detection (**A**) Design of MN array. Reprinted from [81]. Copyright © 2023, with permission from Elsevier. (**B**) Integrated device for regular glucose measurement. Reprinted from [60]. Copyright © 2023, with permission from Elsevier. (**C**) 3D schematic diagram of design structure. Reprinted from [85]. Copyright © 2023, with permission from MDPI. (**D**) CorelDraw rendering of a cross-section of the K^+^ ion-sensitive electrode microfluidic chip. Reprinted from [63]. Copyright © 2023, with permission from John Wiley and Sons, Ltd. (**E**) Schematic representation of the one-touch-activated blood multi-diagnostic system. Reprinted from [66]. Copyright © 2023, with permission from Royal Society of Chemistry. (**F**) Schematic of the proposed minimally invasive blood glucose monitoring system integrating an array of porous MNs in a microfluidic chip. Reprinted from [61]. Copyright © 2023, with permission from Springer. (**G**) Modified and improved system of 4F. Reprinted from [62] Copyright © 2023, with permission from Springer. (**H**) Schematic of an array of horizontal microneedle biosensors in the channel. Reprinted from [65]. Copyright © 2023, with permission from Elsevier.

**Figure 5 pharmaceutics-15-00792-f005:**
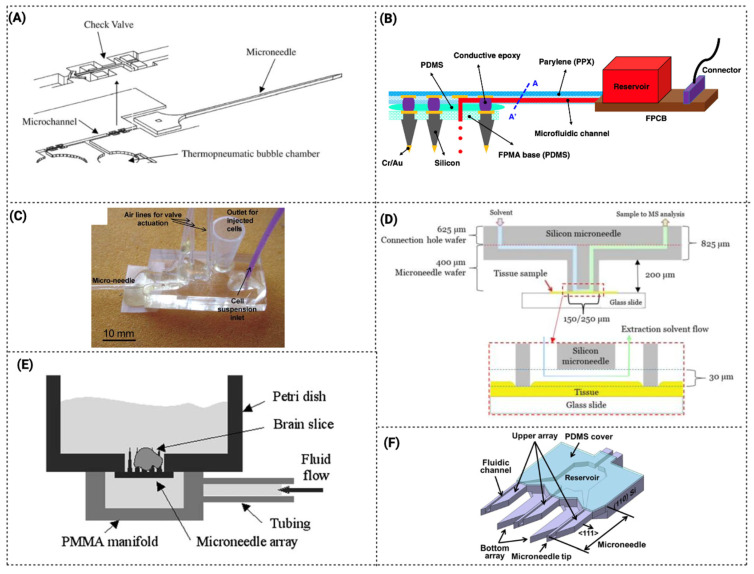
Schematic compilation of MN microfluidic devices for drug delivery and microinjection. (**A**) Schematic of an integrated micropump/microneedle device. The top shows a closeup of a planar free floating directional microvalve. Reprinted from [70]. Copyright © 2023, with permission from Springer. (**B**) The flexible penetrating microelectrode array is integrated with the microfluidic interconnection cable (µFIC). Reprinted from [71]. Copyright © 2023, with permission from Nature. (**C**) Assembled microinjection device. Reprinted from [74]. Copyright © 2023, with permission from Royal Society of Chemistry. (**D**) Silicon microneedle structure showing the solvent injection and sample aspiration during microneedle extraction. Reprinted from [87]. Copyright © 2023, with permission from American Institute of Physics. (**E**) A photomicrograph shows the microneedle array perfusing a brain slice. The microfluidic perfusion system integrates microneedle arrays in a packaged system for fluid containment. Reprinted from [89]. Copyright © 2023, with permission from Springer. (**F**) Schematic illustration of the proposed 2D in-plane microneedle chip. Reprinted from [77]. Copyright © 2023, with permission from Springer.

## Data Availability

Not applicable.

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
