# Peer review of "Microneedles in Advanced Microfluidic Systems: A Systematic Review throughout Lab and Organ-on-a-Chip Applications"

_pharmaceutics, 2023, doi:10.3390/pharmaceutics15030792_

Round 1

Reviewer 1 Report

Authors successfully collect and arrange relevant publications to give readers wide and detailed reviews about microneedles as well as microfluidic systems. Quite lots of publications were checked and introduced in this manuscript. Thus, it is expected that researchers in this field will be very interested with the contents.

However, there are some points to be improved like followings. please refer to the followings.

- Introduction is well-written and easy to understand. It would be better if authors add descriptions about roles of each components, i.e. microneedles and microfluidic systems when they are in one systems, to help readers understand more the importance about integrated system.

- 3.1.1 section is followed by 3.2 section. please check the subheadings and modify them.

- Regarding 3.1.1 and 3.2, authors introduced lots of previous literature. introduction would be OK, however it would be much better to describe the advantages and disadvantages of introduced publications concisely. In addition, it is believed that limitation should exist when two different fields are integrated. Thus, in each section, it is advised to add the description about limitation of research work introduced each section.

- Among references, it seems there are almost same or similar devices with different publications. Please check them, and try to remove abundant one.

- Before conclusion, authors may add descriptions the perspectives in detail as well as what should be necessary or to be improved for future applications. Although this is review manuscript, authors are free to add as much discussion as possible for further breakthrough in this field

Thank you, all authors, for wonderful work.

Reviewer 2 Report

The manuscript titled “Microneedles in advanced microfluidic systems: a systematic review throughout lab and organ-on-a-chip applications” by Renata Maia et. al. summarizes recent advancement in microneedle (MN) systems combined with lab and organ-on-a-chip and specially devotes attention to in the existing types of MN, their fabrication strategy from various materials, and MN applications. Importantly, authors discuss advantages and disadvantages of the MN platform and provide a broad information about potential applications of MNs in microfluidics.

I enjoyed reading this thorough review, the manuscript is organized, well written, and utilizes most recent and relevant publications. The only issue that should be dealt within the manuscript is the overall organization of the review:

Typically, for review-type articles the Materials and Methods as well as Result sections are not necessary. Readers can grasp this information from reading Introduction where a brief paragraph could be added to summarize what methods were used to survey necessary sources.

Reviewer 3 Report

Microneedles (MNs) are a leading novel technology in biomedical field for several applications such as fluid extraction for biomarker detection and delivery of drugs or biomolecules into cells with temporal and spatial precision. MNs can be used not only as a stand-alone tool, but also combined with microfluidic devices. This use has been gained widespread attention for lab-on-a-chip (LoC) devices and recently has created new horizon for drug screening and/or organ models monitoring (organ-on-a-chip (OoC) systems). In this review, the authors provide an overview of MNs applied in sophisticated microfluidic devices (LoC and OoC), showing the recent advancements in these emerging systems. Table 1 summarizes all the important aspects such as MN types and materials, fabrication methods, functions, … The review is well-organized and written comprehensively. No major modifications seem to be necessary for the manuscript, but I recommend accepting it after addressing some issues.

First, I would suggest that the authors summarize the paragraphs 2.1 to 3.1, because in my opinion they detail too much in what way the bibliographical research was conducted.

Line 98-99: I suggest clarifying which surface the authors refer to (for example “fabrication surface”).

Line 272: The authors should report at least one reference for the 3D printing fabrication process.

Line 277: Check the numbering of paragraphs (3.2.1. and not 3.1.1., 3.2.2. and not 3.1.2.)
